# Computerized Ultrasonic Imaging Inspection: From Shallow to Deep Learning

**DOI:** 10.3390/s18113820

**Published:** 2018-11-07

**Authors:** Jiaxing Ye, Shunya Ito, Nobuyuki Toyama

**Affiliations:** National Metrology Institute of Japan (NMIJ), National Institute of Advanced Industrial Science and Technology (AIST), Tsukuba Central 2, Tsukuba 305-8568, Japan; itou-s@aist.go.jp (S.I.); toyama-n@aist.go.jp (N.T.)

**Keywords:** nondestructive evaluation, ultrasonic imaging, computer vision, deep learning, local descriptor, convolutional neural networks

## Abstract

For many decades, ultrasonic imaging inspection has been adopted as a principal method to detect multiple defects, e.g., void and corrosion. However, the data interpretation relies on an inspector’s subjective judgment, thus making the results vulnerable to human error. Nowadays, advanced computer vision techniques reveal new perspectives on the high-level visual understanding of universal tasks. This research aims to develop an efficient automatic ultrasonic image analysis system for nondestructive testing (NDT) using the latest visual information processing technique. To this end, we first established an ultrasonic inspection image dataset containing 6849 ultrasonic scan images with full defect/no-defect annotations. Using the dataset, we performed a comprehensive experimental comparison of various computer vision techniques, including both conventional methods using hand-crafted visual features and the most recent convolutional neural networks (CNN) which generate multiple-layer stacking for representation learning. In the computer vision community, the two groups are referred to as shallow and deep learning, respectively. Experimental results make it clear that the deep learning-enabled system outperformed conventional (shallow) learning schemes by a large margin. We believe this benchmarking could be used as a reference for similar research dealing with automatic defect detection in ultrasonic imaging inspection.

## 1. Introduction

Civil and industrial infrastructures form the backbones of modern society. However, after a long period of service, aging structures, such as pipelines and industrial plants, have become worldwide challenging problems, and undetected structural damages, or a delay in detecting them, can have significant consequences [1]. Nondestructive test (NDT) methods have been developed to detect, locate, and assess damages or flaws in structures without interrupting their continued usefulness or serviceability [2,3]; it is an essential technique to achieving optimal decision-making in the maintenance and rehabilitation of aging structures.

Ultrasonic inspection is a well-established NDT technique for identifying and evaluating internal defects of a wide variety of materials, including metals, plastics, composites, and ceramics [4]. Its general principle is that an electrical pulser is employed to generate an ultrasonic signal that propagates through the inspection object in the form of waves; once a flaw/damage is encountered, part of the wave energy reflects back to the surface of the structure. By investigating the echo waves, the defects are discerned. As a versatile NDT technique, ultrasonic inspection has a couple of favorable merits, such as its high sensitivity to most materials’ damages, and its proficiency in the extraction of defect location and size specifications [5].

With the overwhelming inspection tasks for aging structures, the nondestructive evaluation method is expected to be superior in both efficiency and accuracy. However, current ultrasonic imaging inspection systems are incapable of meeting these two requirements fully. From the efficiency aspect, since the method relies on an inspector’s visual examination to interpret inspection data, the process can become time-consuming as the workload increases. On the other hand, condition assessment is performed in a subjective manner based on an inspector’s individual experience, making the results vulnerable to human errors. In recent years, substantial progress has been made in the computer vision and machine learning research fields, which enable computers to achieve performances that are similar to or even better than humans in multiple visual tasks, such as object detection/recognition [6] and tracking [7]. Aiming to eliminate human errors and efforts in ultrasonic imaging inspection, a new trend is emerging to cast the data interpretation problem in the context of a computer vision technology paradigm [8].

Automatic ultrasonic image data investigation has become an active research theme in recent years. Plenty of efforts have been delivered to design efficient automatic ultrasonic data interpretation systems, and advanced signal processing and machine learning techniques have been adopted to characterize ultrasonic echo waveforms [9] and images [10]. In general, computer-aided ultrasonic data analysis systems consist of four parts: image preprocessing, visual feature extraction, defect pattern identification with statistical machine learning, and final result reporting. In Figure 1, we present a diagram describing the sequential process from ultrasonic signal capture to automatic result generation. Grounded in such fundamental framework, this paper attempts to devise an advanced computer vision system for ultrasonic imaging data interpretation with near-human accuracy. The main contributions of the proposed approach can be summarized as follows.

The objective of this study is to build an efficient automatic ultrasonic image analysis system for NDT. To this end, an ultrasonic inspection image database was established, which consists of 6849 instances. The whole dataset was carefully annotated using binary labels of 0 and 1, denoting the normal and defective case, respectively. We expect to extract critical visual information by using machine learning techniques so as to achieve human-level ultrasonic image understanding.Various state-of-the-art computer vision techniques, including efficient visual descriptors [11] and convolutional neural networks (CNN) [12], have been reviewed and evaluated for the application of ultrasonic echo image pattern classification. Most of them have yet to be applied to the field of ultrasonic image investigation. By presenting side-by-side comparison results, we expect this research can contribute to the field of automatic ultrasonic imaging investigation for NDT by setting a benchmark for future research.It has been acknowledged that, nowadays, the advent of Artificial Intelligence is largely driven by deep neural networks, which enable machines to close the gap to human-level performance for cognition. However, the weak interpretability becomes a significant obstacle in applying deep learning to critical applications [13], i.e., NDT in our focus. That is, no one was sure exactly which features deep learning used to classify an ultrasonic image as healthy or defective. In this study, we conducted extensive experiments to validate convolution neural networks (CNN) for ultrasonic image analysis. Through visualization of the internal representation learned by CNN and in-depth discussion, we demonstrate that the critical visual patterns indicating defective ultrasonic images can be expertly distilled by the neural networks.

## 2. Related Work

The last 5 years have seen the significant advancement of machine learning research and the arisen trend to deploy autonomous systems to relieve people from laborious and exhausting NDT tasks [8,14]. In this section, we present a review of the current research on the subject of automatic ultrasonic data analysis for NDT. The contents are organized with two parts: the first focuses on feature representations adopted for ultrasonic images, and the latter regards investigations of statistical machine learning models for computerized ultrasonic image classification.

### 2.1. Feature Representations for Ultrasonic Signal in NDT

In ultrasonic inspection, the echo signal can be presented in two forms: a waveform and B-scan image. Accordingly, different feature extraction approaches have been employed to convert the (high-dimensional) raw data to a compact form, with the critical information relevant to defect presence well retained. To deal with ultrasonic waves, conventional signal processing methods have been adopted as a mainstream method, such as discrete Fourier transform (DFT) [15] and discrete wavelet transform (DWT) [16]. Subsequently, principal component analysis (PCA) [17] or the genetic algorithm (GA) [18] became commonly applied to eliminate the redundant features further.

As for the case of ultrasonic image characterization, a new series of two-dimensional statistical descriptors have been introduced. For instance, to exploit the texture information of ultrasonic images, summary statistics have been adopted, including auto-correlation; first-/second-order statistical descriptors like skewness, kurtosis, and energy [19]; and the co-occurrence matrix [20]. Some research articles have reported that validated Gabor features can achieve a favorable classification performance with a lab-scale evaluation [21]. However, the efficient visual features broadly used by the computer vision community for complex visual tasks has yet to introduce such methods to the application field of ultrasonic imaging.

### 2.2. Ultrasonic Image Classification Using Statistical Machine Learning

Through the above process of feature extraction, we convert ultrasonic images to a batch of feature vectors; then, statistical machine learning is performed to discern flaw-induced anomalous patterns. In recent years, a wide variety of machine learning techniques have been evaluated for the task, including singular value decomposition (SVD) [22], support vector machines (SVM) [23], and sparse coding (SC) [24]. The literature deems that efficient learning schemes can substantially contribute to ultrasonic data interpretation. It is also noteworthy that neural network (NN)-based learning systems have been repeatedly assessed, from simple to complex structures [25,26], while the neural networks evaluated for the task are limited to four layers so far [6]. The primary factor accounting for the situation is that the ultrasonic inspection datasets are confined to small scales, so there is no apparent performance gain from adopting deep neural networks. It is also unfortunate that there is no standard/public ultrasonic inspection database nor algorithmic benchmark for the application, and, therefore, researchers have to collect the data and perform experiments on their own from scratch.

According to the above survey, this study renders several novel features: first, we introduce a batch of promising visual features from the computer vision community which have never been applied to ultrasonic inspection image analysis; secondly, we adopt the latest deep CNN with far more complex architectures compared to the previous four-layer structures, and it is anticipated to achieve superior performance in discerning subtle defective patterns from ultrasonic images.

## 3. Computer Vision Frameworks under Evaluation

In order to establish an algorithmic benchmark for automatic ultrasonic inspection image analysis, we present a review of computer vision systems and introduce the fundamental techniques in this section. As shown in the upper and lower regions of Figure 2, nowadays, computer vision systems can be generally categorized into two types: the conventional ones which adopt a two-stage pipeline, including hand-crafted feature extraction and statistical machine learning; and another family which relies on deep neural network (DNN) learning architecture to acquire critical feature representation with respect to a given task without human intervention. Driven by the availability of both large-scale annotated datasets and sufficient computation power, the latter DNN-based schemes are taking the place of manual feature engineering in most of the current AI-enabled applications [7]. In our evaluation, we compare both schemes for the focused task of ultrasonic image pattern analysis. We introduce the details of representative techniques as follows.

### 3.1. Conventional Scheme: Visual Feature Extraction + Statistical Classifier

Over the decades, hand-crafted visual features, together with statistical learning-based pattern classification, have long been the standard flow of computer vision systems. Plenty of research attention has been drawn to feature design perfection [27] and classification algorithm induction [28], and promising results have been reported incrementally. However, a comprehensive evaluation in ultrasonic imaging is required. In our benchmarking, we selected the most typical and promising visual features for computerized ultrasonic image investigation.

#### 3.1.1. Feature 1: Local Binary Patterns (LBP)

Local binary patterns (LBP) are effective local descriptors which have been applied for texture classification [29]. LBPs convert local structures into binary patterns by comparing values to the central pixel. In this study, we adopted LBPs as the spectro-temporal feature extractor. The general formulation for an LBP can be written as:(1)$$x_{\mathrm{LBP}}\left(L_{c};\tau_{c}\right)=\sum_{j=1}^{J}\omega_{c}2^{j-1}\left[\;\left[I\left(r_{j}\right)>\tau_{c}\right]\;\right]$$
where I is the gray-level pixel value at the spatial position $r_{j}$, and [[·]] generates 1 only if the bracketed condition is met and 0 otherwise. $L_{c}=\{{r\}}_{j=1}^{J}$ indicates the local 2D structure surrounding $c\in R^{2}$, including *J* spatial positions $r_{j}$ close to c, and $\tau_{c}$ is the gray-level value of the central pixel. For an ordinary LBP, *J* is set to 8 and hence the local patch is limited to 3×3. In this study, we evaluated both a prototype LBP and a modified version, called statistics-based LBP [30]. The latter type employs the following parameters for computing LBPs:(2)$$\tau_{c}=\frac{1}{N}\sum_{i}I\left(r_{i}\right),\;\omega_{c}=\sqrt{\frac{1}{N}\sum_{i}\left(I\left(r_{i}\right)-\mu \right)}$$

Statistics-LBP has been proved to be more robust to additive noises in the images.

#### 3.1.2. Feature 2: Histogram of Oriented Gradients (HOG)

Histogram of oriented gradients (HOG) is one most important 2D local descriptor in computer vision, and it has been proved effective for object detection and identification. The fundamental mechanism of HOG is the count of occurrences of gradient orientations in the localized patch. In this evaluation, we introduced a HOG descriptor to characterize ultrasonic wave patterns. The extracted HOG features, denoted by $x_{HOG}$, are anticipated to be useful for defective pattern detection in ultrasonic images.

#### 3.1.3. Feature 3: Higher-Order Local Auto-Correlations (HLAC)

Higher-order local auto-correlation (HLAC) features are conventional local descriptors for 2D patches [31], and it has been successfully applied to a wide variety of real applications, including texture classification and medical imaging. The mask patterns of HLAC are shown in Figure 3. The HLAC features are well developed based on the higher-order autocorrelation function:(3)$$x_{\mathrm{HLAC}}\left(a_{1},a_{2}\right)=\int S\left(r\right)S\left(r+a_{1}\right)S\left(r+a_{2}\right)$$

In dealing with ultrasonic images, S(r) denotes the input image, $r={\left[t_{r},f_{r}^{\prime }\right]}^{⊤}$ is the reference point on the two-dimension plane, and ($a_{1}={\left[t_{a1},f_{a1}^{\prime }\right]}^{⊤},a_{2}={\left[t_{a2},f_{a2}^{\prime }\right]}^{⊤}$) is a set of displacements. HLAC extraction is limited to a 3×3 local region and, therefore, it is sensitive to local structures, such as ridges and valleys. We introduced a sliding window with three consecutive pixels shifted, then, HLAC features were extracted iteratively. Since ultrasonic echo patterns are assumed to be highly correlated within a local region, more discriminative features can be exploited by HLAC.

#### 3.1.4. Feature 4: Gradient Local Auto-Correlations (GLAC)

The GLAC descriptors are local features based on spatial and orientational auto-correlations of local image gradients [32]. From a geometrical viewpoint, the descriptor exploits information about not only the gradients but also the curvatures of the image surface. Such richer information can contribute to increasing discriminative power than standard histogram-based methods. The GLAC features can be extracted by the following steps: Let I be the ultrasonic image and $r={\left[x,y\right]}^{\prime }$ be the positioning anchor in *I*. The gradient magnitude z and the orientation angle of the image gradient θ at one pixel can be computed by
(4)$$z=\sqrt{\frac{\partial I^{2}}{\partial x}+\frac{\partial I^{2}}{\partial y}},\theta =\arctan\left(\frac{\partial I}{\partial x},\frac{\partial I}{\partial y}\right)$$

Then, the orientation θ is quantized by using *D* discrete bins, resulting in a new feature vector of $f\in R^{D}$. Furthermore, higher-order statistics are extracted as follows.
(5)$$x_{GLAC}=R\left(d_{0},\ldots ,d_{N};a_{1},\ldots ,a_{N}\right)=\int_{I}\omega [z(r,z\left({r+a}_{1}\right),\ldots ,z\left({r+a}_{N}\right)f_{d_{0}}\left(r\right)f_{d_{1}}\left({r+a}_{1}\right),\ldots f_{d_{N}}\left({r+a}_{N}\right)dr$$
where $a_{i}$ are displacement vectors. Based on the auto-correlation function, the GLAC features can be easily extended to explore higher-order statistical features.

These features convert the input high-dimensional pixels to a compact vectorized representation. Then, machine learning algorithms are employed to classify them with minimum errors.

### 3.2. Ultrasonic Image Investigation with Deep Learning

In recent years, artificial neural networks with deep hierarchical architecture (i.e., deep learning) have garnered the most interest due to their superior performance in numerous benchmarking studies of machine perception, such as in speech recognition [33] and image understanding [7].

As the key feature of this research, we customized deep neural networks (DNNs) to discern the defect-induced ultrasonic echo pattern of NDT.

Artificial neural networks, inspired by the neural systems, involve several critical computations of a weighted sum, nonlinear gating, and partial derivatives. Given the data with annotations, DNNs are able to create an efficient mapping between the two modality data. We denote such a function as H(·), which commonly comes up with hierarchical structures. The mechanism of information propagation between layers, e.g., from *k*-1 layer to *k*, complies with the same principal as follows:(6)$$h^{\left(k\right)}=g\left(b^{\left(k\right)}+W^{\left(k\right)}h^{\left(k-1\right)}\right)$$

We denote the DNN structure with stacking layers by *H*. It is noteworthy that g(·), called the activation function, plays a critical role, and we employed the rectified linear unit (ReLU), defined by g(τ,a(k))=max(0,a(k))+a(k)min(0,τ), in this evaluation. We further present an overview of neural network learning in Algorithm 1 [6].
**Algorithm 1:** Train Neural Networks (**x**${}_{t}$,$y_{t}$, **W**${}_{t}$) **Initialization: W**, θ **for**
t=1,2,…,T $\mathrm{do}\left\{\begin{array}{c} \mathrm{Perform}\;\mathrm{forward}\;\mathrm{propagation}:\;{\hat{y}}_{t}=H\left(x_{t},W_{H}\left(\theta \right)\right)\;\\ \mathrm{Compute}\;\mathrm{the}\;\mathrm{prediction}\;\mathrm{loss}:\;L(y_{t},{\hat{y}}_{t})\;\\ \mathrm{Update}\;\mathrm{weights}\;\mathrm{via}\;\mathrm{back}\;\mathrm{propagation}:\;\theta^{t}\leftarrow \theta^{t-1}-\epsilon \frac{\partial L}{\partial \theta }\; \end{array}\right.$ **return**
$W_{H}\left(\theta_{t}\right)$
where $\frac{\partial L}{\partial \theta }$ is the derivative induced by the loss of training data, and ϵ is called the learning rate that governs the network update step/speed. From the diagram, we can see that the DNN iteratively minimizes the prediction error by performing stochastic optimization. Concretely, there are two favorite solvers to tackle the problem: adaptive moment estimation (ADAM) and stochastic gradient descent with momentum (SGDM) [34]. We applied SGDM to our system with a standard mini batch weight updating scheme [6]. Since the objectives are ultrasonic images, we picked convolutional neural networks (CNNs) that are preferred to processing matrix-shaped data. We present two types of CNN designs in this research.

#### 3.2.1. Proposed System 1: USseqNet

We first propose a DNN system with a simple sequential structure, and we introduce the design in Figure 4. Because the layout is not carefully tailored, we treat its performance as the baseline of the deep learning-based approach. In our notations, the input is a 128 × 128-pixel RGB image, and the output is a vector with two entries connected with the softmax classification layer (noted as CL). The proposed scheme consists of four convolution layers, which are C1 ⋯ C4. Batch normalization and ReLU activation gating, noted as B1 ⋯ B4 and R1 ⋯ R4, respectively, are applied after the convolution operation. P3 is a pooling layer to reduce the size of the feature map. D4 indicates the dropout layer which can effectively prevent overfitting by discarding a ratio of neural connections [34]. At the fully connected (FC) layer, we flatten all weights and obtain a high-dimensional vector representation. Finally, the classification layer (CL) converts the data value to defect/no-defect categorical membership score.

#### 3.2.2. Proposed System 2: USresNet

A wide variety of CNN frameworks have been developed for visual tasks throughout the decades by alternating the number of layers, filter shapes, layer types, and connection paths between layers [6]. Among them, well-developed CNNs, such as AlexNet, VGG-16, and ResNet architectures, emerged as standard approaches to visual tasks [35]. Extensive empirical and theoretical research results demonstrate that deeper neural nets are anticipated to be superior in distilling critical patterns from data for a given task. However, as the neural network hierarchy goes deep, the error back-propagates through the network and the gradient shrinks, thus affecting the ability to update the weights in the earlier layers for deep networks [34]. To tackle such a “vanishing gradient problem”, Residual net (ResNet) was proposed, which introduces a “short cut” module which contains an identity connection such that the weights can directly propagate to very early layers. The short cut module learns the residual mapping by a formula of H(x)=F(x)+x [35]. Taking inspiration from ResNet, we propose a deeper CNN design compared to previously applied systems for ultrasonic image pattern analysis. There are two features of our proposal. In the first place, more layers were adopted, which enables the extraction of rich discriminant information to discern defect/no-defect in ultrasonic images; secondly, by employing the “short cut” module, the weight updating can be performed smoothly for the whole network. We present the proposed “USresNet” architecture in Figure 5 and the experimental validation results in Section 4.

We explain the detailed process at each layer as follows. The input and output are a 128 × 128-pixel RGB image and a two-element vector of defect/no-defect membership scores, respectively. There are seven convolution layers employed, noted as C1 ⋯ C7. At each layer, the convolution operations are performed to seek for specific defect-induced ultrasonic wave patterns. The convolution kernel size is set to 3 × 3 pixels. Batch normalization and ReLU activation gating, noted as B1 ⋯ B7 and R1 ⋯ R7, are applied after convolutions. Notably, three "short cut" modules are adopted, that is, by assigning adding layers noted as A1 ⋯ A3, back-propagating weights can jump ahead to earlier layers without decay. Re1 ⋯ Re3 denotes the ReLU gating performed at the combination layer of each residual module. Right before the fully connected layer (FC), a drop layer Dr3 is employed to prevent overfitting. Finally, the classification layer (CL) computes the probability of defect presence by using the softmax function.

## 4. Experimental Validations

### 4.1. Ultrasonic Propagation Image Data Collection

In this section, we introduce the ultrasonic inspection image dataset we created to evaluate computer vision algorithms. To collect the imaging data, we employed the ultrasonic imaging inspection system described in Ref. [5], which consists of three parts: a pulsed laser scan unit which generates thermoelastic ultrasonic waves, a transducer attached to the surface of a specimen that collects the ultrasonic waves propagated through the structure, and an amplifier and a digital oscilloscope (A/D converter) that transmits the captured echo signal to a computer, where the data is stored on the hard disk drive. A snapshot of the traveling waves at a given time is obtained by plotting the amplitude of each waveform at that time on a contour map using the reciprocity principle in wave propagation [36]. The snapshots are continually displayed in a time series and appear as a video of traveling waves generated from the fixed receiver. The key parameters applied in the laser ultrasonic imaging system are presented in Table 1.

The shape, size, and depth of flaws are critical parameters affecting visual inspection. To evaluate the automatic ultrasonic image investigation approaches, we prepared a batch of stainless steel plates with various types of flaws. Table 2 shows the specimen details and Figure 6 presents our ultrasonic inspection schemes. In the evaluation, two types of defects were investigated: drill holes with a diameter from 1 to 5 mm and slits with lengths ranging from 3 to 10 mm. A laser scan is performed on the central region of 3 mm thick specimens with a 100 × 100 mm size (green zone) on both the front and back sides of the steel plates. It is noteworthy that the incident angle of the transducer installation can be a critical parameter that significantly affects the ultrasonic imaging patterns, especially when examining slit flaws. To exploit the robustness of computer vision algorithms to those variations, we collected ultrasonic inspection images with different incident angles varying from 0 to 90 with a 22.5 interval.

As a result, we captured 6849 ultrasound images with a 3235/3069 split between no-defect and with-defects cases. In Figure 7, we present the sample patches from both cases. People can discern the defective images at a glance by focusing on the flaw-induced echoes. However, for algorithms, it is very difficult to define such echo signals with explicit programming. Through this work, we evaluated state-of-the-art computer vision algorithms for this challenging task.

It is noteworthy that data augmentation was an important trick applied in the evaluation; it contributes to preventing the model from overfitting the dataset. Deep CNN models are featured with superior ability to “memorize” the training data [34], that is, it can always achieve near perfect accuracy at the training stage. However, when new data arrive with a broader variance, the performance can degrade severely. Various data augmentation methods were considered to enhance the generalization power of the learning model, such as cropping, translation, and rotation. In this evaluation, we performed extensive data augmentation. Table 3 and Figure 8 show some augmented samples. The dataset laid the fundamentals for further numerical analysis.

### 4.2. Experimental Settings

This section demonstrates the parameters used in the evaluation. At the feature extraction stage shown in Section 3, the computation of HOG is limited to the local region with 8 × 8 pixels, while HLAC, LBP, and GLAC are extracted from 3 × 3 grids. By such settings, two-dimensional local variants can be characterized. In addition to dealing with individual visual features, we added an option to concatenate the four types of feature values as one *long* feature vector, which we call “fusion” in the evaluation. To classify the feature vectors, we employed support vector machine (SVM) with the Gaussian kernel. The spread parameter is determined at the validation stage. Dropout ratios at both the D4 stage in USseqNet and Dr3 stage in USresNet were set to 0.5. The mini-batch size was set to 256 for USseqNet and 64 for USresNet. We limited the updating epochs to 10. We set the initial learning rate to 1.0 × 10${}^{-3}$, and it was scheduled to drop 10% for every 20 iterations. As suggested in many previous studies, we set the Momentum parameter to 0.9 in the stochastic gradient descent optimizer. At the evaluation stage, we adopted a leave-one-specimen-out protocol. By using the settings shown in Table 1, we collected ultrasonic scan images from 18 specimens, including 17 defective samples with various flaw specifications presented, as in Table 2, and one plate without defects. To conduct an unbiased evaluation, we tested the images collected from one specimen at each iteration, and the images captured from the other specimen were used to train the model. As iteration goes, images collected by every specimen can be tested. Finally, we obtained the predicted labels for the whole dataset.

### 4.3. Empirical Evaluation Results

In this part, we present results obtained from experimental validation. First, we introduce the evaluating metrics, which are an essential factor for assessing the prediction models. By comparing the prediction results with ground truth labels, we can derive four major statistics:True Positive (TP): number of defect images correctly detected.True Negative (TN): number of normal images classified as no-defect.False Positive (FP): number of normal images incorrectly detected to have defects.False Negative (FN): number of defect images incorrectly classified as no-defect.

On the basis of these, we introduce four metrics, i.e., precision (Pr), Recall (Re), Accuracy (ψ), and F-score (γ):(7)$$Pr=\frac{TP}{TP+FP},\;Re=\frac{TP}{TP+FN},\;\psi =\frac{TP+TN}{P+N},\;\gamma =2\cdot \frac{Pr\cdot Re}{Pr+Re}$$

According to the result summary in Table 4, we highlight several key findings. First of all, the comparison of accuracies and F-scores indicate that the proposed deep learning-based USresNet achieved the best performance for ultrasonic image investigation. The contributions of the deep architecture and efficient “short cut” link module were clarified by comparing the USresNet to USseqNet which obtained F-scores of 94.47% and 93.33%, respectively. Secondly, we compare the two families of computer vision systems, which are the conventional methods that rely on feature engineering with statistical classification, and the ones based on feature learning by using neural networks. When comparing the well-designed GLAC features with baseline sequential CNN model of USseqNet, we can see the results are very close. That means that even using a quite simple structure, deep learning systems can compete with top-tier hand-crafted visual features, in other words, deep learning schemes already become an off-the-shelf tool for ultrasonic image analysis. Moreover, we found that fusion of multiple visual features can attain performance gain. Such a scheme can also work to combine conventional visual features with learned representations from neural networks. In addition to the above image-based evaluation, we further provide specimen-based results in Figure 9. The underlying assumption is that the quality of ultrasonic inspection images can vary according to the property of flaws. We hope the computerized ultrasonic image analysis system can be robust to those variations.

Figure 9 reveals more details regarding the evaluation, as the detection accuracies for each specimen are shown. It is evident that the defect-induced patterns from several specimens are more difficult to distinguish, such as those from No. 1–6. In general, slit defects are easier to detect from computerized visual inspection compared to the hole flaws. Notably, conventional visual features are incapable of characterizing critical discriminant information for flaw detection, and severe accuracy drops can be seen when dealing with hole damage. On the contrary, deep learning-based solutions are quite robust. Especially for the USresNet design, there is quite a small performance variance for all the specimens. As a result, the proposed USresNet outperformed all other methods in both accuracy and stability.

Though the performance of USresNet is superior to that of the others, it is not perfect in the sense of statistics, compared to manual labels. To find the reasons, we examined the misjudged images and show a subset of them in Figure 10. The images are annotated as with-defect-presence, while the USresNet regarded them as no-defect. The characteristic feature among these images is that the (central) defective regions are small, and there are no defect-generated echoes shown. The misdetection issue can be attributed to the size of the defects. Since the labels are assigned to each ultrasonic image, USresNet draws more attention to the global patterns than to the localized structures. One possible solution to this issue is to segment the image into small crops and refine the labels for subregions. We leave this for future work.

### 4.4. Data and Model Visualization

Figure 11 shows the visualization results, where the binary class labels are annotated with different colors. It is evident that deep learning effectively maps the raw images to new space, where the class-wise discriminant power is enhanced. The visualization agrees with the evaluation results.

Data and model visualization are acknowledged as an integral part of machine learning systems: they make complex data properties more accessible and render a better understanding of the established model. In this research, we incorporated a data visualization function with two objectives. The first is to understand the massive raw data distributions by using a standard dimension reduction method of principal component analysis (PCA) with t-distributed stochastic neighbor embedding (t-SNE) [6]. Then, we illustrate the data residing in the new feature space created by deep learning. By comparing the two distributions, the contribution of task-driven deep learning for feature extraction can be validated. In the experiment, we extracted the full connected (FC) layer outputs of USresNet as neural projection data and visualized them with t-SNE.

In addition to seeing the global data distribution, we need to access more details about the learned neural network model; that is, from the raw pixels of the image, the kind of information that has been characterized by the deep learning pipeline. To this end, we visualize both the front-end (C1 layer in USresNet) and intermediate representations (C4 layer in USresNet) in Figure 12 and Figure 13, respectively. Those illustrations deem that the hierarchical neural networks can characterize the input data through decomposition and reconstruction. Compared with hand-crafted visual features, the deep neural networks with multiple stacking-layers are anticipated to be more robust and efficient in dealing with variational input data.

Finally, we investigate the activation of intermediate layers in the USresNet. Deep learning systems usually consist of billions of weights, and they are well structured to detect specific patterns from image pixels. Activation visualization is a useful approach to interpret the convolution neural networks by exploiting what the neuron “is looking for” [37]. If the pattern appearing in the image is sensitive to the neural network, high activation scores will be generated. We present a demonstration in Figure 14, including both input image and the corresponding activation at the B4 layer of USresNet. It is evident that the flaw-induced round-shape echoes generated high activation scores while, on the contrary, the ultrasonic wave propagation mode was suppressed. The visualization confirms that the USresNet establishes very effective visual pattern analysis pipeline for defect detection in ultrasonic images.

## 5. Discussion and Conclusions

The development of automatic data interpretation with high efficiency and accuracy for nondestructive evaluation has emerged as an active research theme in recent years. In this paper, we attempt to develop an efficient computerized system for ultrasonic image investigation with comparable accuracy to human. To this end, we first established an ultrasonic inspection image dataset, including more than 6000 annotated instances. Then, we performed a comparative evaluation of computer vision techniques for automatic ultrasonic image pattern investigation for NDT. The candidate methods range from conventional styles with hand-crafted features and statistical classifiers to the latest deep learning systems. To the best knowledge of the authors, this is the first study in ultrasonic inspection imaging testing with an extensive evaluation of representative computer vision methods. The experimental comparison validated that the proposed USresNet with deep neural networks outperformed all conventional approaches by a large margin in both detection accuracy and model stability. Also, to obtain a better understanding regarding the learned USresNet, we analyzed its behavior in feature mapping. The results confirm that the USresNet can mimic humans to capture round-shaped defect-induced echoes for anomaly detection in ultrasonic images. In a further research, we hope to deal with subtle defect pattern detection from ultrasonic images. Currently, the system can only tell whether a defect exists, and it is not able to report further information, such as location and size. Those will be our further topics. Furthermore, it can be anticipated that as we expand our ultrasonic image dataset, the analysis precision can be further improved. 

## Figures and Tables

**Figure 1 sensors-18-03820-f001:**
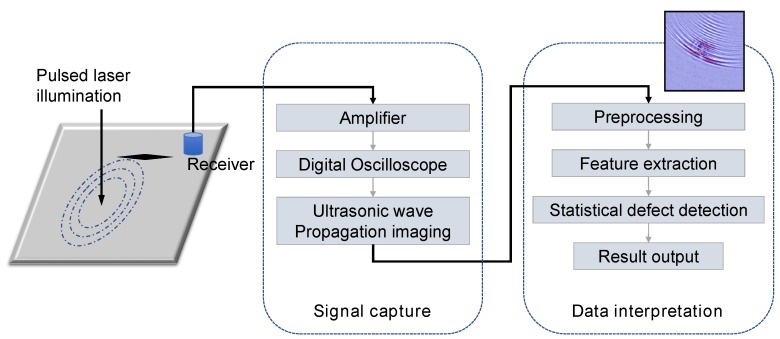
The general processing flow of computerized ultrasonic imaging inspection.

**Figure 2 sensors-18-03820-f002:**
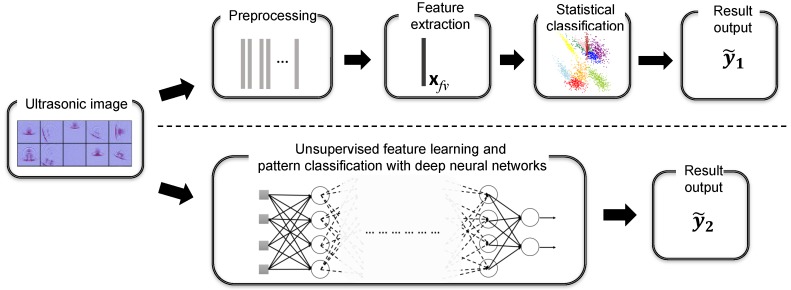
The general designs of computer vision systems.

**Figure 3 sensors-18-03820-f003:**
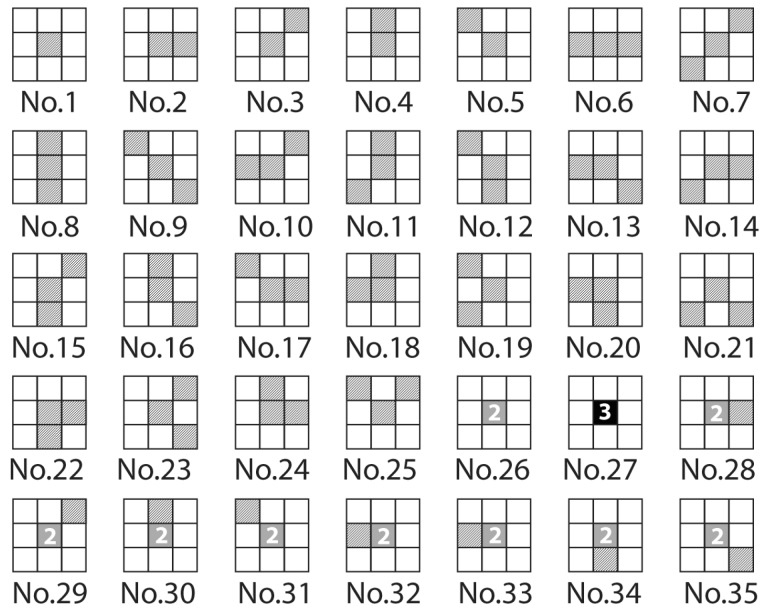
Feature mask patterns of HLAC.

**Figure 4 sensors-18-03820-f004:**
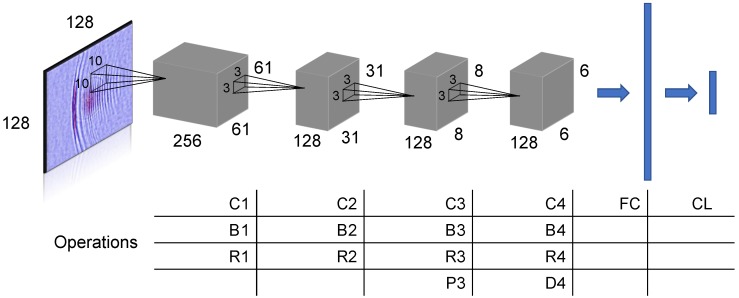
The proposed USseqNet architecture.

**Figure 5 sensors-18-03820-f005:**
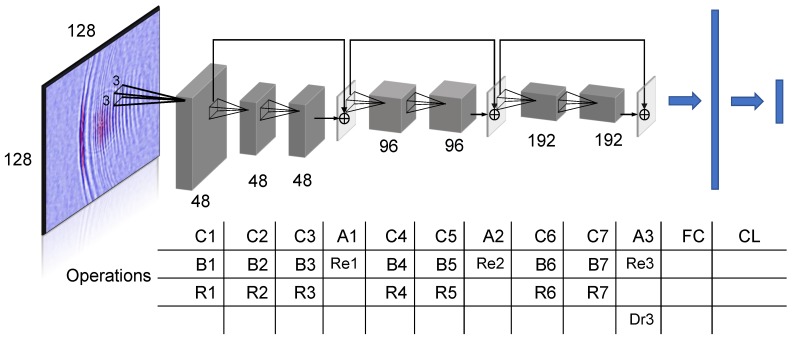
The proposed USresNet architecture.

**Figure 6 sensors-18-03820-f006:**
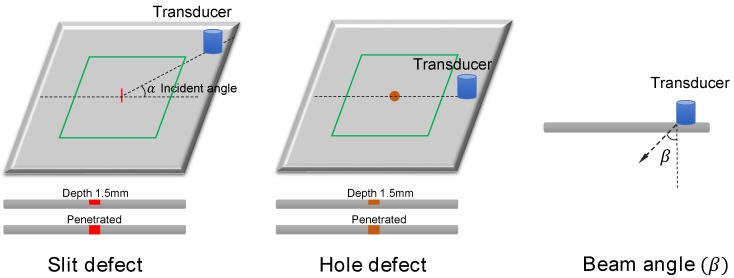
Sample patches from the dataset, including the ones with defects (**left**) and no defects (**right**).

**Figure 7 sensors-18-03820-f007:**
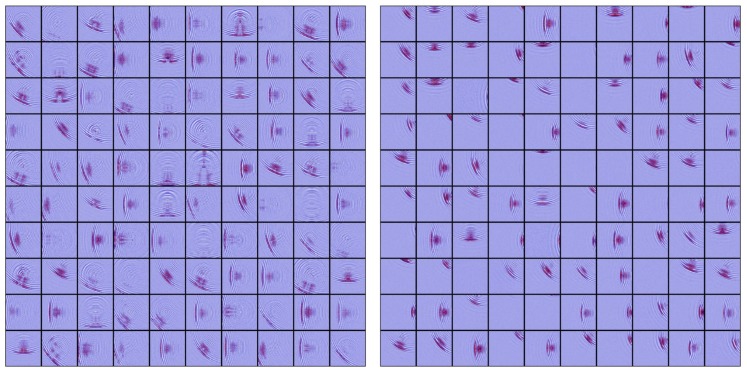
Sample patches from the dataset, including the ones with defects (**left**) and no defects (**right**).

**Figure 8 sensors-18-03820-f008:**
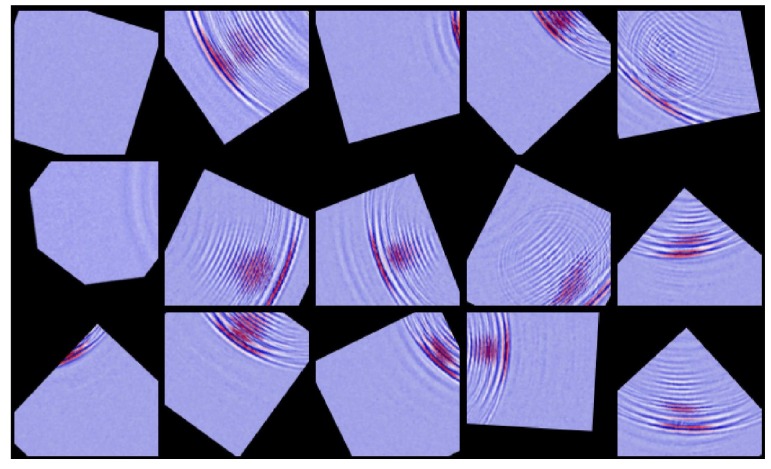
Sample patches from augmentation.

**Figure 9 sensors-18-03820-f009:**
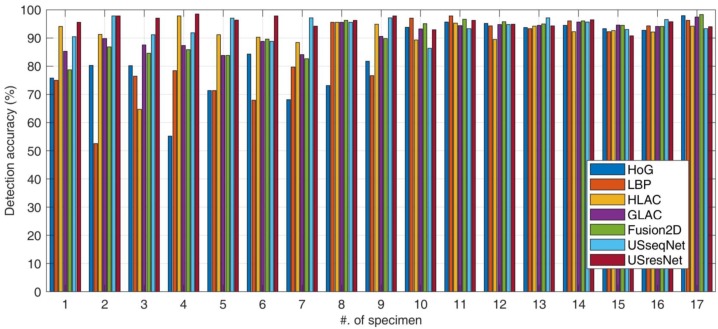
Specimen-wise classification accuracy comparison.

**Figure 10 sensors-18-03820-f010:**
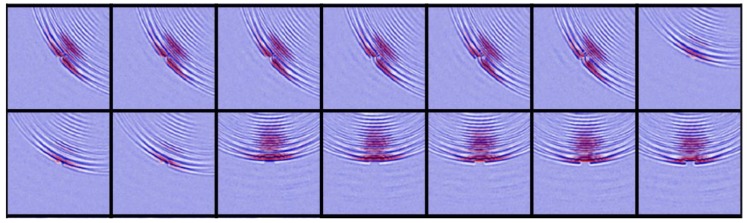
Sample images of misclassification.

**Figure 11 sensors-18-03820-f011:**
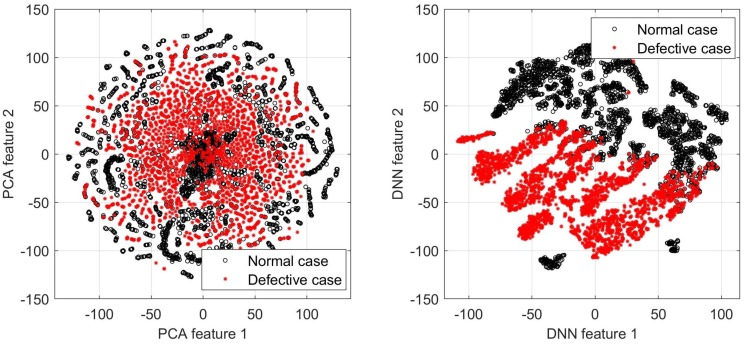
Comparison of the feature space established by raw image and convolutional neural network (CNN).

**Figure 12 sensors-18-03820-f012:**
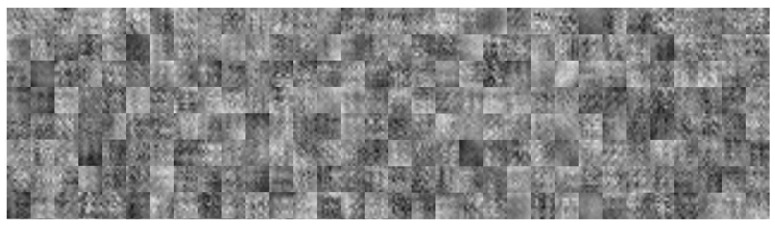
Visualization of C1 layer weights learned by USresNet.

**Figure 13 sensors-18-03820-f013:**
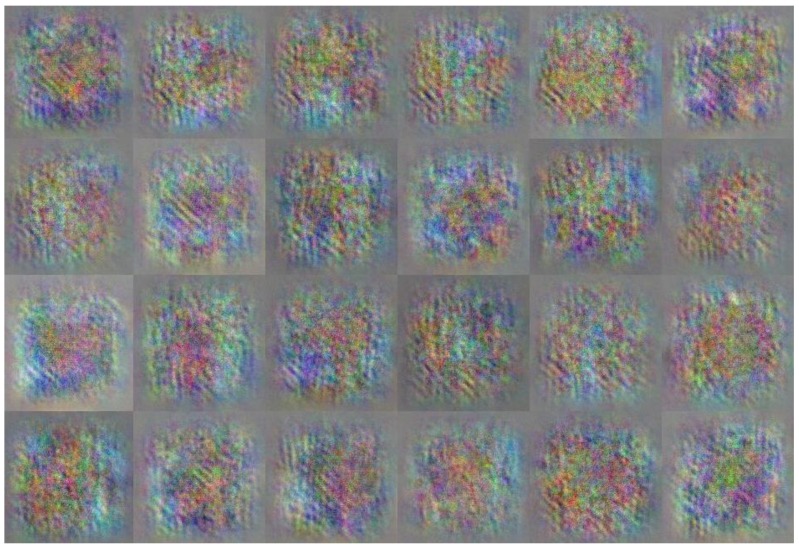
Visualization of C4 layer weights of USresNet.

**Figure 14 sensors-18-03820-f014:**
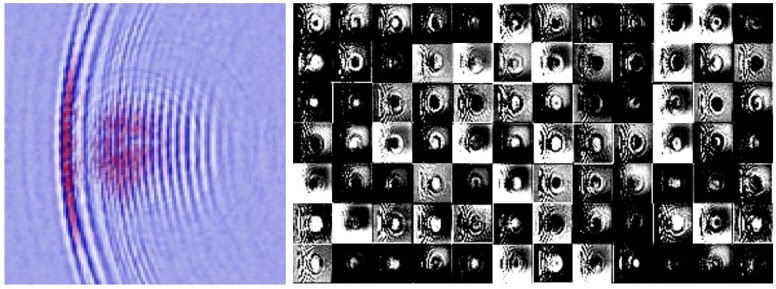
Visualization of USresNet B4 layer activation generated by an ultrasonic image with defect.

**Table 1 sensors-18-03820-t001:** Summary of the experimental setting of inspection device.

	Item	Setting
1	Probe frequency	1 MHz
2	Beam angle	90
3	Pulse repetition frequency	500 Hz
4	Incident angle α(°)	0, 22.5, 45, 67.5, 90

**Table 2 sensors-18-03820-t002:** Summary of the specimen specifications.

Specimen #.	Flaw Type	Depth	Transducer Side	Defect Size
1–3	Hole	Penetrated	Front	ϕ 1 mm | ϕ 3 mm | ϕ 5 mm
4–6	Hole	1.5 mm	Front	ϕ 1 mm | ϕ 3 mm | ϕ 5 mm
7–9	Hole	1.5 mm	Back	ϕ 1 mm | ϕ 3 mm | ϕ 5 mm
10–11	Slit	Penetrated	Front	5 mm | 10 mm
12–14	Slit	1.5 mm	Front	3 mm | 5 mm | 10 mm
15–17	Slit	1.5 mm	Back	3 mm | 5 mm | 10 mm

**Table 3 sensors-18-03820-t003:** Data augmentation scheme.

	Type	Parameter Sample Range
1	Reflection	*X*-axis, *Y*-axis
2	Rotation angle	[−20, 20] degrees
3	Scaling	[0.5, 1]
4	Translation	X: [−50,10], Y: [−10,50]

**Table 4 sensors-18-03820-t004:** Results of automatic ultrasonic image pattern classification using various methods. The bold figures denote optimal results under evaluation criteria.

	HoG	LBP	HLAC	GLAC	Fusion2D	USseqNet	USresNet
Precision (%)	89.27	89.95	91.43	**96.10**	95.98	95.28	93.98
Recall (%)	92.70	92.73	92.44	90.75	91.82	91.46	**95.67**
Accuracy (%)	90.80	91.20	91.90	93.54	94.00	93.48	**94.73**
F-score (%)	90.95	91.32	91.93	93.35	93.86	93.33	**94.47**

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
