# Peer review of "Computerized Ultrasonic Imaging Inspection: From Shallow to Deep Learning"

_sensors, 2018, doi:10.3390/s18113820_

Reviewer 1 Report

The authors developed shallow and deep learning approach for automatic ultrasonic nondestructive testing from the obtained ultrasonic images. Such algorithm and approach could save the labor cost and enhance the inspection accuracy. This work is valuable for this field.

It is never mentioned that how to form the ultrasonic image.

Detection performance using the features as shown in Figure 11 should also be included. It is preferred to compare the shallow and deep learning performance.

The minimum size of slit and hole defects should be investigated.

What are the influences of ultrasound parameters, such as central frequency, signal-to-noise ratio?

Discussion of the influence of different specimen material and geometry should also be included.

Abstract 1-3 This statement is not very accurate. First, ultrasonic wave propagation cannot be visualized. Second, ultrasonic NDT does not only rely on the observation of images, but also the analysis of waveforms.

Page 2 Line 10 bad use of “as well”

Page 6 Line –7 grammar error

Page 7 Line 16 grammar error

            Line 19  change “convert” to “converts”

Fig. 5 What do the operations of ‘A’, ’Dr’, and ‘Re’ mean?

Page 8 Line 10-11 hard to understand this sentence

            Specification of lasers should be included

Page 12 Line 5 bad use of “vary to”

Page 14 Line -1 hard to understand this sentence

Page 15 Line 5 there are no data of human accuracy in this paper. So the claim of “comparable accuracy to human” is not supported.

          Line 18  bad use of “also”

Author Response

We sincerely thank the reviewer for valuable comments and modification suggestions, which were of great help in improving the manuscript. Accordingly, the revised manuscript has been broadly improved. Our responses [Author's comments  (AC)] to the referee’s comments (RC) are given below.

[RC] It is never mentioned that how to form the ultrasonic image.

[AC] Thank you for the comments regarding the ultrasonic imaging data collection. Since we had been focusing on the algorithm development, this part was missed. Sorry for that. Now, we have added the related information:

A snapshot of the traveling waves at a given time is obtained by plotting the amplitude of each waveform at that time on a contour map using the reciprocity principle in wave propagation [36]. The snapshots are continually displayed in a time series and appear as a video of traveling waves generated from the fixed receiver.

This device had been developed by my research group. For more details, please refer to the paper

[36] Yashiro, S., Takatsubo, J., Miyauchi, H., Toyama, N. A novel technique for visualizing ultrasonic waves in general solid media by pulsed laser scan. NDT \& E International, 2008 41(2), 137-144.

Thank you for your attention.

[RC] Detection performance using the features as shown in Figure 11 should also be included. It is preferred to compare the shallow and deep learning performance.

[AC] Let us present more details. In Figure 11, the right-hand image showing the ultrasonic image data distribution in the feature space learnt by USresNet neural network. Actually, we already use those feature representations to classify the ultrasonic images. In detail, Those features were extracted at FC(fully connected) layer in the flow chart of Figure. 5. And in the next step CL layer (classification layer), we fed statistical classification model (softmax) to those features and establish a optimal classifier classify black/red points which denote defect-free ultrasonic images and the ones with flaw. The quantitative evaluation results are shown in both Table 4 (USresNet column) and Figure 9 (USresNet).

[RC] The minimum size of slit and hole defects should be investigated.

[AC] In our current evaluation, the minimum size of defects are 3 mm slit and 1 mm diameter hole. According to the analysis results, those defects can be detected. It is a pity that our current database doesn't cover specimen with even smaller defects. We totally agree to the comment that the ultimate defection ability is critical issue and we plan to further expand our dataset, to collect inspection images from smaller defects and other shapes. It will be one key topic in the future work.

[RC] What are the influences of ultrasound parameters, such as central frequency, signal-to-noise ratio? Discussion of the influence of different specimen material and geometry should also be included.

[AC] Thank you for the comment. Yes, the same as in conventional ultrasonic inspection, the central frequency of the transducer is the major parameter. In this study, the parameters we focus on the image pattern analysis with clean images, therefore we choose the parameters which ensure the signal-to-noise-ratio of the data is sufficiently high. More detailed discussions, such as hardware system settings and variations of specimen,  will be our major topic in future works.

[RC ]Abstract 1-3 This statement is not very accurate. First, ultrasonic wave propagation cannot be visualized. Second, ultrasonic NDT does not only rely on the observation of images, but also the analysis of waveforms.

[AC] Thank you for the valuable comment. We had revised the expression to make the contents more correct. Ultrasonic imaging is very common technique but visualization of wave propagation is indeed not. We thank the review to point out this fundamental issue.

[RC] Page 2   Line 10  bad use of “as well”

Page 6 Line –7 grammar error

Page 7 Line 16 grammar error

             Line 19  change “convert” to “converts”

Fig. 5 What do the operations of ‘A’, ’Dr’, and ‘Re’ mean?

Page 8 Line 10-11 hard to understand this sentence

            Specification of lasers should be included

Page 12 Line 5 bad use of “vary to”

Page 14 Line -1 hard to understand this sentence

Page 15 Line 5 there are no data of human accuracy in this paper. So the claim of “comparable accuracy to human” is not supported.

               Line 18  bad use of “also”

[AC] All above issues regarding writing/typos had been addressed in the updated manuscript. The English and grammatical errors are corrected and the expressions have been significantly improved. We thank the review for careful reviewing of the manuscript.

So far, all comments presented by reviewer had been responded. We hope the current version can be well-suited to be presented to the readers of Sensor Journal.

Reviewer 2 Report

The manuscript entitled “Computerized Ultrasonic Imaging Inspection: from Shallow to Deep Learning”, with the following ID sensors-384955, presents the development of a methodology for an automatic and highly accurate data interpretation for non-destructive evaluations  based on ultrasonic image investigation. Additionally, the authors present an extensive evaluation of representative computer vision methods for this specific application. Overall, this paper presents a well performed and extensive literature review. The methodology is also good and the results interesting. The results are well discussed and good insights for futures works are presented. Nevertheless, there are some issues that the authors might consider reviewing:

- Please consider reviewing the paper in terms of English language. Overall, there are some problems with verbs and regular tenses.

- Please consider the deletion of repetitive acronyms description. One example is CNN for instance in the introduction. There are others in the manuscript.

- Please review some wrongly used capital letters (Examples: ”Aforementioned Feature” or “NOT” in section 2.2).

- Please check Figure 4. It looks like there is a problem with P4<->B4

- In table 1, some units are missing and should all of them should be at the same column for the sake of coherence.

- Please consider spacing between numbers and units. Examples “3mm”. The same applies for other forms of text “…90°with 22.5°interval.”

- Please check the word “epochs”.

- The authors state “According to the Tab.1, the ultrasonic scan images have been collected from 18 specimens with different flaws specifications.” It seems the authors might want to refer to Tab. 2 and not 1. Also, in Tab.2, 17 specimens are presented, not 18.

Author Response

We sincerely thank the reviewer for valuable comments and modification suggestions, which were of great help in modifying the manuscript. Accordingly, the revised manuscript has been broadly improved. Our responses to the referee’s comments (RC) are given below.

Comments upon language and typos:
- Please consider reviewing the paper in terms of English language. Overall, there are some problems with verbs and regular tenses.
- Please consider the deletion of repetitive acronyms description. One example is CNN for instance in the introduction. There are others in the manuscript.
- Please review some wrongly used capital letters (Examples: ”Aforementioned Feature” or “NOT” in section 2.2).
- Please check Figure 4. It looks like there is a problem with P4<->B4
- In table 1, some units are missing and should all of them should be at the same column for the sake of coherence.
- Please consider spacing between numbers and units. Examples “3mm”. The same applies for other forms of text “…90°with 22.5°interval.”
- Please check the word “epochs”.

All above issues had been addressed in the updated version of manuscript. We carefully examined the English and grammatical errors and the readability had been significantly improved.

- The authors state “According to the Tab.1, the ultrasonic scan images have been collected from 18 specimens with different flaws specifications.” It seems the authors might want to refer to Tab. 2 and not 1. Also, in Tab.2, 17 specimens are presented, not 18.

We agree to reviewer on this point and made revision on the part explaining our experiment settings. Tab. 1 shows the device setting and Tab.2 presents the test specimen specifications. Now the expression becomes more clear. As for the Tab. 2, those 17 specimens are with flaws and we further prepared one specimen without defect. Adding them up, there are 18 specimen under evaluation. We are sorry for the unclear expression in last version and now this part has been presented clearly.

So far, all comments had been addressed and we hope the current version can be well-suited to present to readers.